# Simulation and Optimization: A New Direction in Supercritical Technology Based Nanomedicine

**DOI:** 10.3390/bioengineering10121404

**Published:** 2023-12-08

**Authors:** Yulan Huang, Yating Zheng, Xiaowei Lu, Yang Zhao, Da Zhou, Yang Zhang, Gang Liu

**Affiliations:** 1State Key Laboratory of Vaccines for Infectious Diseases, Xiang An Biomedicine Laboratory, National Innovation Platform for Industry-Education Integration in Vaccine Research, State Key Laboratory of Molecular Vaccinology and Molecular Diagnostics, Center for Molecular Imaging and Translational Medicine, School of Public Health, Xiamen University, Xiamen 361102, China; huangyulan@stu.xmu.edu.cn (Y.H.); zhengyt980926@163.com (Y.Z.); gangliu.cmitm@xmu.edu.cn (G.L.); 2Institute of Artificial Intelligence, Xiamen University, Xiamen 361002, China; luxiaowei@stu.xmu.edu.cn; 3Shenzhen Research Institute, Xiamen University, Shenzhen 518000, China; zhaoy@xmu.edu.cn; 4School of Mathematical Sciences, Xiamen University, Xiamen 361005, China

**Keywords:** supercritical fluids, model, machine learning, computational fluid dynamics, particles

## Abstract

In recent years, nanomedicines prepared using supercritical technology have garnered widespread research attention due to their inherent attributes, including structural stability, high bioavailability, and commendable safety profiles. The preparation of these nanomedicines relies upon drug solubility and mixing efficiency within supercritical fluids (SCFs). Solubility is closely intertwined with operational parameters such as temperature and pressure while mixing efficiency is influenced not only by operational conditions but also by the shape and dimensions of the nozzle. Due to the special conditions of supercriticality, these parameters are difficult to measure directly, thus presenting significant challenges for the preparation and optimization of nanomedicines. Mathematical models can, to a certain extent, prognosticate solubility, while simulation models can visualize mixing efficiency during experimental procedures, offering novel avenues for advancing supercritical nanomedicines. Consequently, within the framework of this endeavor, we embark on an extensive review encompassing the application of mathematical models, artificial intelligence (AI) methodologies, and computational fluid dynamics (CFD) techniques within the medical domain of supercritical technology. We undertake the synthesis and discourse of methodologies for calculating drug solubility in SCFs, as well as the influence of operational conditions and experimental apparatus upon the outcomes of nanomedicine preparation using supercritical technology. Through this comprehensive review, we elucidate the implementation procedures and commonly employed models of diverse methodologies, juxtaposing the merits and demerits of these models. Furthermore, we assert the dependability of employing models to compute drug solubility in SCFs and simulate the experimental processes, with the capability to serve as valuable tools for aiding and optimizing experiments, as well as providing guidance in the selection of appropriate operational conditions. This, in turn, fosters innovative avenues for the development of supercritical pharmaceuticals.

## 1. Introduction

According to current definitions, nanomedicines are nanoscale tools utilized for disease diagnosis, prevention, and treatment [1,2]. They have the capacity to facilitate early disease detection and prevention, direct a bioactive molecule to the intended site of action, and regulate the release of the molecule to guarantee an ideal concentration at the target of therapy for the intended duration [3,4].

Numerous nanomedicines, including liposomes, metal-organic frameworks (MOFs), polymeric nanosystems, and magnetic nanoparticles, offer advantages over traditional therapies that generally alter the pharmacokinetics and pharmacodynamics (pK/pD) of the active ingredients [5,6]. However, there are still a number of issues with traditional nanosizing techniques including precipitation, emulsion, and spray-drying that need to be resolved [7].

Supercritical fluid (SCF) technology has recently demonstrated significant promise in the biomedical field due to its advantages of being eco-friendly, sustainable, safe, and non-toxic. It can be used for the extraction of active ingredients from natural medicines, the synthesis and reaction of drugs, and the preparation of nanomedicines [8]. The application of SCF technology to the preparation of nanopharmaceutical particles provides a simpler and better-controlled process for the development and production of nano- and micro-sized particles and is easily adaptable to the concepts of green chemistry and green engineering [9]. This technique involves mixing a drug solution with an SCF (e.g., SCCO_2_, etc.) and ejecting it from a tiny aperture nozzle, which results in the formation of solid nanoparticles with the rapid vaporization of the SCF. Due to its simple process, low cost, and use of environmentally friendly solvents, which enable the preparation of high-purity nanocrystals without any organic solvent residue, SCF technology has attracted the interest of a wide range of researchers in comparison to traditional methods of preparing nanoparticles [10]. For example, Liu’s group applied the SCCO_2_ antisolvent precipitation method to directly nanocrystallize the clinically used drugs doxorubicin (DOX) [11] and indocyanine green (ICG) [12] and mixed the nanoparticles with lipiodol by simple ultrasonication to obtain the super-stable homogeneous lipiodol-drug formulations. The results indicated that there were no appreciable structural or functional differences between the original drugs and nanoparticles. Additionally, the formulations had a superb slow-release effect that improved the effectiveness of interventional embolization in hepatocellular carcinoma and had good clinical application value [13].

Despite having excellent application potential in the field of nanomedicine, the complexity of the equipment and engineering, consistency and quality control, and control of operational parameters are a few of the problems and obstacles that SCF technology faces. Taking the preparation of pure pharmaceutical nanoparticles by SCF technology as an example, the characteristics of the particles, such as yield, size, and morphology, depend on the process conditions employed, including pressure, temperature, solvent type, flow rate, and the design of the nozzle and collection chamber [14]. For instance, research has been performed on how temperature and pressure affect particle size and product composition [15]. According to the findings, astaxanthin content in co-precipitated particles decreased as the working pressure under a constant temperature was raised from 8 MPa to 15 MPa, and both particle size and astaxanthin content increased as the working temperature under constant pressure was raised from 40 °C to 60 °C. The particle size reduced when the Polyvinylpyrrolidone (PVP) ratio of astaxanthin was increased from 5:1 to 20:1, and the ratio of 10:1 (at 60 °C and 10 MPa) was determined to have the maximum astaxanthin content in the co-precipitates. When creating drug particles using SCF technology, two factors interact to produce particles with vastly different sizes, shapes, and morphologies: first, the drug’s solubility in the SCCO_2_, and second, the mixing of the organic solution with the SCCO_2_. The size of the medication particle can be altered in accordance with solubility. In general, high solubility favors the rapid expansion method, i.e., the particle formation process in which SCCO_2_ is used as a solvent, which causes a higher rate of nucleation due to a higher supersaturation of the solute in SCCO_2_. The increase in the rate of nucleation causes the formation of a large number of crystals, which reduces the average particle size [16]. On the other hand, low solubility favors a supercritical anti-solvent (SAS)-based particle formation process [17]. Therefore, optimizing the preparation conditions of nanomedicines is a complex and tedious task.

However, due to the special characteristics of supercritical equipment, the reactor is sealed to maintain a high-temperature and high-pressure environment, which makes it impossible to measure the process parameters directly in real time. Changing the process parameters frequently to determine the ideal conditions in the actual situation is not only a waste of time but also requires a significant amount of research and development funds and production resources. Therefore, interdisciplinary collaboration is required to address the aforementioned issues as well as the challenges associated with the widespread use of SCF technology in the biomedical industry. Many predictive models for studying the preparation of nanoparticles from SCCO_2_, including empirical and semi-empirical models, equations of state (EoS), and machine learning (ML), can predict the solubility of drugs in SCCO_2_ [18]. The capacity to generate continuous solubility values at various temperatures and pressures from discrete experimental data to investigate the best operating conditions is the major benefit of such predictive solubility models over experimental measurements. This review summarizes computational models using mathematical models including semi-empirical models and EoS, AI methods, and computational fluid dynamics (CFD) techniques applied in SCF technology for pharmaceuticals (Figure 1). The implementation process and results of these models are compared to explore how to select simple but effective predictive models in light of different situations and to provide theoretical approaches to solving the bottlenecks in SCF technology.

## 2. Mathematical Models

In this review, mathematical models refer to models constructed based on mathematical methods and theoretical foundations to describe and explain phenomena through the formulation of mathematical equations. These models utilize equation-solving techniques to predict outcomes under various experimental conditions. Supercritical nanoparticle formation techniques commonly employ empirical models, semi-empirical models, and EoS to forecast drug solubility in SCCO_2_. Empirical and semi-empirical models can fit a range of solubility values under different operating conditions by using a discrete set of experimental data. The EoS, grounded in thermodynamic principles, is a physically derived model describing the relationship between pressure, temperature, and density. Consequently, based on its theoretical underpinnings, it can effectively predict solubility beyond the range of experimental data. Furthermore, molecular dynamics models can be employed to ascertain the transport properties of nanoparticles produced through supercritical methods, providing insights into mixing efficiency during the supercritical particle formation process.

### 2.1. Empirical Models

The utilization of SCF technology represents one of the most effective approaches for reducing side effects and enhancing bioavailability by reducing the particle size of drugs to the micro or nanoscale. The solubility of drugs in SCCO_2_ constitutes a crucial factor for the successful preparation of particles, particularly those of small dimensions. Similar to the presence of crossover points, it should be noted that temperature and pressure do not necessarily exert singular and stable influences on solubility. As illustrated in Figure 1, pressure exhibits a unidirectional effect on the solubility of drugs in SCCO_2_, with higher pressures resulting in increased solubility (Figure 1a). Conversely, the impact of temperature on solubility is intricate due to the existence of a “crossover pressure” at specific pressure values. When operating conditions align with this particular pressure, solubility remains relatively constant irrespective of temperature variations (Figure 1b). When the pressure is below the crossover pressure, the reduction in density predominantly governs the solubility of drugs in SCCO_2_, with solubility decreasing as the temperature rises. Conversely, when the pressure exceeds the crossover pressure, the sublimation pressure takes precedence in influencing solubility, resulting in an increase in solubility with increasing temperature [23]. In experiments conducted by Alshahrani et al., the solubility of metoprolol in SCCO_2_ was measured to be approximately 0.45 × 10^−5^ mol·mol^−1^ at a crossover pressure of 180 bar [24]. The solubility of tamoxifen, a cancer-fighting medication, in SCCO_2_ was measured to be approximately 1.0 × 10^−4^ mol·mol^−1^ under the conditions of a crossover pressure of 200 bar by Pishnamazi and colleagues [25]. Hence, to investigate the influence of operational parameters on the solubility of drugs in SCCO_2_, it is necessary to separately explore the impacts of temperature, pressure, and density as well as the combined effects of temperature, pressure, and density. Experimental determination of supercritical solubility under various operating conditions is a time-consuming, expensive, and challenging endeavor. Moreover, relying solely on experimental measurements of drug solubility yields discrete data points, making it difficult to discern the continuous variation trends in solubility across different operating conditions. Therefore, empirical models that do not involve physical fundamentals or critical properties are typically considered the preferred approach to correlate the solubility of different substances in SCCO_2_ under varying conditions.

Empirical models do not rely on detailed physical principles or equations of a system; instead, they summarize, generalize, and fit experimental data to mathematical expressions used for describing and predicting system behavior. Empirical models are often preferred when addressing practical problems due to their simplicity, lack of stringent requirements on compound properties, and high correlation accuracy. These straightforward empirical models, derived solely through fitting experimental data, may at times exhibit strong alignment between modeling and measurement data. However, such empirical equations might only be applicable to specific situations, and they may lack persuasiveness when extrapolated beyond the existing range of experimental data. Consequently, they may not be suitable for similar experiments with different materials or under different conditions.

Semi-empirical models occupy a middle ground between empirical models and physical models. They rely on physical principles, necessitating the construction of some physical equations, followed by parameter optimization through the fitting and adjustment of experimental data [26]. Due to the high cost and complexity associated with experimentally measuring the solubility of drugs in SCCO_2_ over a wide range of temperatures and pressures, utilizing measured experimental data to fit and optimize the parameters of semi-empirical models becomes imperative. Once the precision falls within an acceptable range, these models not only enable the correlation of solubility as a function of temperature and pressure but also facilitate the inference of drug solubility and the prediction of pressures and temperatures beyond the scope of experimental investigation [27].

Empirical and semi-empirical models are derived from observed changes in solubility data, and they employ mathematical formulas to express trends in solubility. Consequently, by fitting discrete experimental data points, these models allow the construction of continuous curves that represent the impact of temperature, pressure, and density on solubility. This facilitates the prediction of solubility under unmeasured conditions and the exploration of optimal operating conditions within a certain range. Improvements to empirical and semi-empirical models primarily involve the addition of parameters to existing models from prior research, enhancing their precision in representing solubility changes. As depicted in Figure 2, six commonly used empirical models were employed to predict the solubility of Febuxostat in SCCO_2_. Solubility is defined as a function of SCCO_2_ density and temperature, and in the fourth group of models (iv), the influence of pressure on solubility was also considered. The discrete data points represent experimentally measured data, while the curves represent the calculated results obtained by fitting the experimental data using empirical models. It is evident that the experimental measurement points align closely with the curves generated by these six empirical models, all exhibiting a similar trend to the experimental data. To assess the accuracy of different models, the Average Absolute Relative Deviation (AARD%) was employed. AARD% is a standard for assessing the disparity between experimental and calculated values. A lower AARD% indicates a smaller deviation between the model and experimental data. Among the six models considered, the Keshmiri model (vi) exhibits the lowest AARD% = 10.630, signifying the highest fitting accuracy [28]. The specific forms and characteristics of these six empirical models are outlined as follows:

Chrastil’s model (1982), based on the concept of chemical association and the formation of chemical complexes between solvents and solute molecules in organic solvents, revealed through experimental data that the logarithm of solute solubility is linearly related to the logarithm of the density of SCCO_2_. The consideration of pressure’s impact on solubility is indirect, as it links solute solubility to the density and temperature of the pure solvent (1). The primary advantage of this model is its reliance on only the density and temperature of the SCF, along with experimental solubility data [29], where *y*_2_ is the solubility of the drug in SCCO_2_, $\rho_{1}$ is the density of SCCO_2_, ai are adjustable parameters and *i* = 1, 2, 3…, and *T* is the temperature of the system.
$$y_{2}=\frac{{\left(\rho_{1}\right)}^{a_{1}-1}\exp\left(a_{2}+\frac{a_{3}}{T}\right)}{\left[1+{\left(\rho_{1}\right)}^{a_{1}-1}\exp\left(a_{2}+\frac{a_{3}}{T}\right)\right]}$$
(1)$$Iny_{2}=a_{1}+a_{2}In\rho_{1}+\frac{a_{3}}{T}$$

Bartle’s model (1991) (2) involves pressure as a linear function of density, where *P* is the pressure, incorporating Pref as the reference pressure and $\rho_{ref}$ as the reference density. This inclusion allows for the correlation to encompass terms related to the correction of the temperature’s impact on the corrected enhancement factor (*y*_2_
*P*)/*P_ref_* [30].
(2)$$In\left(\frac{y_{2}P}{P_{ref}}\right)=a_{1}+a_{2}\left(\rho_{1}-\rho_{ref}\right)+\frac{a_{3}}{T}$$

Sung and Shim’s model (1999) (3) is a modification of the Chrastil model (1). It introduces a relationship between temperature and density, resulting in solubility isotherms exhibiting a linear correlation with fluid density or the logarithm of density [31].
(3)$$Iny_{2}={(a}_{1}+\frac{a_{2}}{T})In\rho_{1}+\frac{a_{3}}{T}+a_{4}$$

Hozhabr’s model (2014) (4) is a model that introduces a new constant after rearranging the Mendenz-Santiago and Teja (MST) model (5) [32]. The MST model (2000), generally abbreviated as the MST model, is based on dilute solution theory. Research on this model revealed that solubility data for binary systems can be plotted on a straight line over a substantial range of temperatures and pressures. Consequently, the equation obtained after fitting the parameters can be used to predict solid solubility under conditions where no measurements have been made. Importantly, this equation is often utilized to assess the consistency of data [33].
(4)$$TIn\left(y_{2}P\right)=a_{1}+a_{2}\rho_{1}+a_{3}T$$
(5)$$Iny_{2}=a_{1}+\frac{a_{2}}{T}+\frac{a_{3}\rho_{1}}{T}-a_{4}InP$$

The Adachi and Lu model (1983) (6) is an extended version of the Chrastil model (1). It involves the correction of the coefficient of Inρ1 and takes into account the relationship between density and the association number [34].
(6)$$Iny_{2}=a_{1}+{(a}_{2}+a_{3}In\rho_{1}+a_{4}In\rho_{1}^{2})In\rho_{1}+\frac{a_{5}}{T}$$

The Keshmiri model (2014) (7) is based on the observation of a linear relationship between Iny_2_ and Inρ1 under certain temperature and pressure conditions. Additionally, it acknowledges the non-linear relationship of Iny_2_ with temperature and pressure. Consequently, this empirical equation was proposed [35].
(7)$$Iny_{2}=a_{1}+\frac{a_{2}}{T}+a_{3}P^{2}+\left(a_{4}+\frac{a_{5}}{T}\right)In\rho_{1}$$

A unified accuracy standard, the “Average Absolute Relative Deviation (*AARD*%)”, is employed to compare the precision of different models, with the *AARD*% calculation defined as follows (8):(8)$$AARD\%=\frac{1}{N}\sum_{i=1}^{n}\left|\frac{y_{2,i}^{Cal}-y_{2,i}^{exp}}{y_{2,i}^{exp}}\right|\times 100\%$$

Here, *y*_2,_*_i_^Cal^* represents the solubility calculated by the model, *y*_2,_*_i_^exp^* denotes the experimental solubility, and N represents the number of experimental data points. A smaller *AARD*% indicates higher model accuracy.

When selecting empirical and semi-empirical models to explore the optimal solubility of drugs in SCCO_2_, the accuracy of these models is associated with the number of parameters. Generally, a greater number of adjustable parameters leads to better-fitting results. Table 1 presents empirical and semi-empirical models used in the past five years for calculating drug solubility in SCCO_2_ [36,37,38]. These 32 models are categorized into five groups based on influencing factors (variables): density; temperature and density; temperature and pressure, density; temperature and pressure; and pressure and density. Sodeifian et al. utilized 20 models, which varied in the number of parameters from 3 to 6, to correlate the solubility of Dasatinib monohydrate (DAS) in SCCO_2_. Among these models, Yu et al.’s model with six adjustable parameters exhibited the best performance [39]. Sajadian et al. employed 30 empirical and semi-empirical models to predict the solubility of Lenalidomide (LND) in SCCO_2_. Among these models, the one proposed by Stahl et al., which incorporates only one variable (density) and two parameters, exhibited the lowest accuracy with an AARD% as high as 47.73% [40].

In these 32 models, the most frequently employed ones are those based on temperature and density functions, including the Chrastil model (11 instances), the K-J model (7 instances), the Sung and Shim model (6 instances), the Garlapati-Madras model (6 instances), and the Adachi and Lu model (5 instances). Additionally, models based on temperature, pressure, and density have been utilized, including the Bartle model (12 instances), the MST model (9 instances), the Keshmiri model (5 instances), and the Jouyban model (5 instances).

The accuracy of fitting 12 drug solubilities for these 9 models is compared, as presented in Table 2. It can be observed that the Bartle model and the Chrastil model exhibit lower accuracy, both of which have only three adjustable parameters. In contrast, the Adachi and Lu model and the Keshmiri model, each with five adjustable parameters, demonstrate relatively better accuracy. The higher accuracy of the Kumar-Johnston (K-J) model can be attributed to the fact that this study only compared it with models containing 3–4 parameters for tamoxifen, Chlorothiazide [19], and Lacosamide [41]. Similarly, in the investigation of Chlorothiazide solubility in SCCO_2_, the Jouyban model, with six adjustable parameters, outperforms other models in terms of accuracy [43].

### 2.2. EoS-Based Models

The method of deriving result data from equations constructed by a physical model, without the need for fitting parameters using experimental data, is referred to as a physics-based model. In the realm of SCF technology, commonly employed physical models encompass thermodynamic models and molecular dynamics models. Thermodynamic models are well-suited for macroscopic-scale descriptions of the thermodynamic behavior and phase transitions of a system, particularly when it involves variations in temperature and pressure. They can be used to calculate drug solubility, density, and phase behavior in SCCO_2_ [44,45,46].

Another method for calculating the solubility of solids in SCCO_2_ involves the use of solid-liquid equilibrium models and EoS. Based on the thermodynamic equilibrium definition between gas and solid phases, the system is divided into an SCF phase (component 1) and a solid phase (component 2). The solubility of the solute in the SCF phase can be determined by equating the fugacity of the solute in the SCF phase to the fugacity of the solute in the solid phase under equilibrium conditions (9). This approach assumes certain conditions: that carbon dioxide does not dissolve in the solid phase, that the purity of the solid solute remains constant, and that the molar volume of the solute is independent of pressure. These conditions are used to calculate the solubility (*y*_2_) of the drug in SCCO_2_ [47,48].
(9)$$f_{2}^{SCF}=f_{2}^{solid}$$
(10)$$y_{2}=\frac{P_{2}^{sub}\exp\left(\frac{V_{s}\left(P-P_{2}^{sub}\right)}{RT}\right)}{{\hat{\phi }}_{2}^{SCF}P}$$
where $f_{2}^{SCF}$ is the fugacity of the solute in the SCF phase, $f_{2}^{solid}$ is the fugacity of the solute in the solid phase, $P_{2}^{sub}$ is the sublimation vapor pressure of the solid, *V_s_* is the molar volume of the solid, and ${\hat{\phi }}_{2}^{SCF}$ is the fugacity coefficient of the solid in the SCF phase.

The fugacity coefficient is determined through an EoS. Similar to empirical models, EoS models can yield continuous curves to explore patterns of solubility variation. Commonly employed EoS models are presented in Table 3.

As illustrated in Figure 3, EoS models are utilized to compute the solubility of the drug Chloroquine in SCCO_2_, with discrete points representing experimental measurements. In both (a) PR-EoS and (b) SRK-EoS models, the discrete points closely align with the curves, confirming the high accuracy of various EoS models. The correlation indicators R^2^ for both models exceed 0.9, indicating that these models can explain over 90% of the variability. Hence, the dissolution of chloroquine in SCCO_2_ can be predicted using these models. Therefore, the optimal EoS model may vary for different drugs and requires specific analysis. The Peng–Robinson (PR) equation remains the most widely utilized EoS model for addressing supercritical issues [20,49,50,51].

However, the process of supercritical nanoparticle formation involves more than just one substance, namely SCCO_2_. Therefore, when using an EoS, it is necessary to employ mixing rules to describe the behavior of mixed systems accurately. The impact of mixing rules on accuracy is evident. The vdW2 outperforms the other three mixing rules (Figure 4a), and the mrPR also demonstrates good accuracy (Figure 4b). Mixing rules with two interaction parameters, such as the vdW2 and the mrPR, exhibit higher accuracy compared to mixing rules with only one parameter [52]. The specific expressions for mixing rules are as follows, where a, b, and c are parameters for models, *y_i_* is the molar mass of the I component, and *k_ij_*, *l_ij_*, and *m_ij_* are the binary interaction parameters in the mixing rules:
Van der Waals-1 parameter mixing rule (vdW1) (11), with one parameter, *k_ij_*:$$a=\sum_{i}\sum_{j}y_{i}y_{j}a_{ij};a_{ij}=\sqrt{a_{i}a_{j}}\left(1-k_{ij}\right)$$
$$b=\sum_{i}\sum_{j}y_{i}y_{j}b_{ij};b_{ij}=\frac{b_{i}+b_{j}}{2}$$
(11)$$c=\sum_{i}\sum_{j}y_{i}y_{j}c_{ij};c_{ij}=\frac{c_{i}+c_{j}}{2}$$Van der Waals-2 parameters mixing rule (vdW2) (12), with two parameters, *k_ij_* and *l_ij_*:$$a=\sum_{i}\sum_{j}y_{i}y_{j}a_{ij};a_{ij}=\sqrt{a_{i}a_{j}}\left(1-k_{ij}\right)$$
$$b=\sum_{i}\sum_{j}y_{i}y_{j}b_{ij};b_{ij}=\frac{b_{i}+b_{j}}{2}\left(1-l_{ij}\right)$$
(12)$$c=\sum_{i}\sum_{j}y_{i}y_{j}c_{ij};c_{ij}=\frac{c_{i}+c_{j}}{2}$$Panagiotopoulos–Reid mixing rule (mrPR) (13), with two parameters, *k_ij_* and *k_ji_*:$$a=\sum_{i}\sum_{j}y_{i}y_{j}a_{ij};a_{ij}=\sqrt{a_{i}a_{j}}\left[1-k_{ij}+\left(k_{ij}-k_{ji}\right)y_{i}\right]$$
$$b=\sum_{i}\sum_{j}y_{i}y_{j}b_{ij};b_{ij}=\frac{b_{i}+b_{j}}{2}$$
(13)$$c=\sum_{i}\sum_{j}y_{i}y_{j}c_{ij};c_{ij}=\frac{c_{i}+c_{j}}{2}$$Mukhopadhyay–Rao mixing rule (MR) (14), with one parameter, *m_ij_*:$$a=\sum_{i}\sum_{j}y_{i}y_{j}a_{ij}{\left(\frac{b}{b_{ij}}\right)}^{m_{ij}};a_{ij}=\sqrt{a_{i}a_{j}}$$
$$b=\sum_{i}y_{i}b_{i};b_{ij}=\sqrt{b_{i}b_{j}}$$
(14)$$c=\sum_{i}y_{i}c_{i};c_{ij}=\sqrt{c_{i}c_{j}}$$

Compared to empirical models and semi-empirical models, the use of an EoS model yields higher accuracy in calculating solubility [53]. Sodeifian et al. employed empirical models, a new solid phase-liquid phase equilibrium model, and the PR- vdW2mr to compute the solubility of Sulfabenzamide (an antibacterial drug) in SCCO_2_, and they observed similar trends. The EoS models, particularly the PR equation, exhibited the highest level of correlation accuracy [43]. This is attributed to the fact that empirical models typically only consider temperature, pressure, and density, whereas EoS models necessitate the consideration of physicochemical properties and critical characteristics, involving a more extensive and comprehensive set of parameters. Most critical parameters can be obtained through reference sources, with binary interaction parameters expressed as linear relationships between temperature and other variables, necessitating fitting to experimental data; setting these binary interaction parameters to zero would result in decreased accuracy [54].

In addition to examining drug solubility, the transport properties of drugs in SCCO_2_ are also of great significance, and mathematical models can be employed to address these aspects. EoS models are capable of calculating the density of SCF and mixed solutions. Molecular dynamics models are suitable for describing molecular interactions at a microscopic scale, enabling the study of transport properties in SCF technology, such as viscosity, thermal conductivity, and diffusivity, by simulating the motion trajectories of a large number of molecules [55,56,57]. Santos et al. utilized a molecular dynamics model to predict the diffusion coefficients of alcohols in SCCO_2_. They compared the predictive capabilities of the model with experimental results, revealing consistent trends [58]. For entirely novel systems where information is entirely lacking, molecular dynamics models can prove to be effective tools for computing solid-state diffusion rates.

Empirical models are readily accessible, but they lack a solid physical foundation and are ill-suited for complex physical systems. Moreover, their accuracy is limited, particularly when extrapolating results beyond the experimental range, making extrapolation unreliable. When investigating the impact of pressure and temperature on drug solubility in SCCO_2_, the preferred approach is to initially employ empirical models for rapid curve fitting, especially when direct solubility data measurement is feasible. It is advisable to select empirical models with a greater number of adjustable parameters to ensure accuracy. Physical models, such as thermodynamic models, fall short in terms of microscale accuracy and fail to provide detailed substance motion information. In contrast, molecular dynamics models offer higher precision in describing microstate conditions but come with the trade-off of complexity and labor-intensive computational processes. In cases where experimental data are limited, employing physical models to calculate drug solubility, density, viscosity, thermal conductivity, and other properties in SCCO_2_ is a preferable choice.

Regardless of the model type, it is impossible to assess the precision of mathematical models when data are lacking. However, it is feasible to compare the computational outcomes of different models, delineate disparities among them, and elucidate potential influencing factors. Each of the aforementioned mathematical models has its own limitations, which should be considered in context. Combining these models, where appropriate, can lead to a more comprehensive and accurate analysis and prediction of problem-solving. In engineering research, acquiring experimental data for SCF can be cost-prohibitive. Instead, researchers can utilize the NIST REFPROP database to access the properties of SCCO_2_ and mixed solutions [59,60]. The NIST REFPROP software version 10.0 (NIST, Boulder, CO, USA) program stands as a powerful tool for calculating the thermophysical properties of industrially significant fluids. Within its capabilities, numerous reliable models are incorporated, enabling the direct retrieval of properties such as density and viscosity for both pure fluids and their mixtures [61].

## 3. AI Models

In recent years, with advancements in technology, AI has rapidly developed. Due to its inherent qualities such as efficiency, cost-effectiveness, and time-saving capabilities, it has found applications across various domains and gained significant attention from numerous experts and scholars. By constructing different AI models, various tasks like clustering, classification, regression, and more can be accomplished. These models can provide solutions to problems in different fields and scenarios, serving as valuable references or auxiliary tools. Furthermore, due to the rapid problem-solving capabilities of AI, it can greatly reduce the demands on human resources, physical resources, and even financial resources. This plays a pivotal role in the dissemination and promotion of AI [62].

Currently, the technology of SCF has found widespread applications across various domains of production and life, including drug particle manufacturing and the extraction of natural organic compounds. In the pharmaceutical industry, in particular, there has been a continuous surge in the preparation of pharmaceutical formulations based on SCF technology. This approach enables the micronization of drug particles or enhances their solubility and stability. Simultaneously, the integration of AI techniques into the realm of SCF pharmaceutical formulation preparation has gained significant attention. The fundamental concept involves fitting AI models using existing experimental data to establish regression models for solubility data within specific temperature and pressure ranges. This, in turn, facilitates the prediction of various parameters related to drug formulations. Such an approach can substantially reduce experimental time and costs, providing indispensable assistance to researchers. Furthermore, it brings new opportunities for revolution and innovation in the field of medicine.

However, acquiring effective data in pharmaceutical formulation preparation proves to be challenging and time-consuming. Often, the recorded useful data during the preparation experiments for each drug formulation are both limited and scarce [63,64]. This is incongruent with the current advancements in big data and AI technologies. The data generated during pharmaceutical formulation preparation are insufficient to train a large-scale ML model that exhibits a satisfactory fitting performance. Consequently, in numerous studies, researchers have resorted to utilizing classical ML algorithms to address tasks related to SCF-based pharmaceutical formulation preparation. Remarkably, these approaches have yielded commendable predictive results [65,66,67,68]. Conversely, employing models with a high number of parameters may lead to overfitting of the data, resulting in less-than-ideal predictive outcomes.

Using AI techniques for result prediction involves several key steps. Initially, a substantial dataset is essential, serving as the training set. The model undergoes training while fine-tuning its parameters, allowing it to effectively capture and learn the relationships within the data. When the parameter values reach a point of minimal change, it signifies the completion of model training. Subsequently, the trained model is employed to predict outcomes based on other variable values. In contrast to mathematical models, AI methods offer a more intuitive representation of data fluctuations. As illustrated in Figure 5i, the three-dimensional model depicts the solubility data of tamoxifen in SCCO_2_ with pressure and temperature as axes. Employing AI methodologies, including Decision Trees (DT), Adaptive Boosting Decision Trees (ADA-DT), and Nu-Support Vector Regression (Nu-SVR), enables the prediction of tamoxifen solubility in SCCO_2_ [21]. The fundamental concepts and characteristics of these three models are outlined below:

DT is a fundamental and commonly used data analysis model capable of handling regression or classification tasks. The feature space is divided into several units, each with a specific output. For test data, we simply assign them to a particular unit based on their features to obtain the corresponding output value. Therefore, it can be employed to predict the solubility of drugs in SCCO_2_ under different temperatures and pressures. As depicted in Figure 5ii, a typical DT comprises decision nodes, edges, and terminal or leaf nodes. The process of partitioning units is also the process of constructing the tree. With each partition, the corresponding output for the unit is determined, effectively adding a new node to the tree.

The ADA-DT model optimizes the DT model through the AdaBoost technique. In each iteration, AdaBoost sequentially trains a series of weak classifiers (DT). The process of iterative training adjusts the weights of the DT until a predefined number of iterations or performance requirements are met. Ultimately, the prediction result is obtained by summing the weighted outputs of all DTs, creating a strong classifier.

The Nu-SVR model aims to compute the correlation as shown in Equation (15), primarily focusing on optimizing tightness and flatness. *ϕ*(*x*) represents a non-linear function that maps the input space to a higher-dimensional space, where o denotes the bias and *w^T^* represents the weight vector.
(15)$$f\left(x\right)=w^{T}\phi \left(x\right)+o$$

To assess the performance of different models in regression tasks, several commonly used metrics are employed, as shown in Table 4. Among these metrics, R^2^, representing the coefficient of determination that measures the extent to which the independent variables explain the variance in the dependent variable, is of particular significance. A higher R^2^ value indicates a better model fit, approaching unity. Additional metrics include the Mean Squared Error (MSE), Root Mean Squared Error (RMSE), Mean Absolute Percentage Error (MAE), Mean Absolute Percentage Error (MAPE), and Mean Absolute Percentage Error (Max Error). These metrics collectively reflect the disparities between the fitted data and actual data, with smaller values indicating superior model fitting. As depicted in Figure 5iii, a numerical comparison was conducted between the actual results and model estimations for the solubility of tamoxifen in SCCO_2_ using the three models. The horizontal axis represents experimental values, while the vertical axis represents predicted values. The majority of training points closely align with the diagonal, signifying their proximity to the actual values. Based on the correlation coefficient R^2^ between the models and experimental data, the R^2^ values for the DT model, ADA-DT model, and Nu-SVR model are 0.836, 0.921, and 0.813, respectively. Consequently, the ADA-DT model, with an R^2^ exceeding 92.1%, demonstrates the highest explanatory power and is deemed the most representative of the experimental data, while the Nu-SVR model exhibits the lowest precision.

In this study, a statistical analysis of pharmaceutical data utilized in the prediction of drug formulation solubility through AI methods has been conducted, as presented in Table 5. Presently, the majority of pharmaceutical applications involve the use of supercritical solvents, predominantly SCCO_2_. When applying AI techniques, a selective approach has been employed, wherein only a subset of experimental data have been utilized. This subset comprises variables that are not only comprehensive but also exert the most significant influence on the outcomes. These variables encompass parameters such as pressure, temperature, SCCO_2_ density, drug solubility, and drug density. This approach serves a dual purpose: it facilitates data cleansing and reduces redundancy while enabling the model to further unearth the intricate relationships between important variables and drug solubility. It is noteworthy that the variables affecting the solubility of different drugs, along with their respective value ranges, exhibit variations, and experimental data quantities also vary accordingly.

Once the dataset has been established, the next step involves selecting models that are appropriate for the respective dataset. In the context of solubility prediction for pharmaceutical formulations based on SCF processes, the objective is to predict the solubility of drug formulations given specific values of variables such as supercritical solvent temperature and pressure. This task falls into the category of regression tasks within the realm of AI. Table 6 provides a statistical overview of commonly employed ML regression models in this context [69,70,71].

In recent years, significant progress has been made in the fields of AI-SCF technology and nanomedicine, owing to the substantial advancements in computer hardware that have greatly enhanced computational capabilities. In the realm of SCF technology, AI models are not only applied to predict drug solubility but also assist in optimizing experimental design and parameters [72]. In the medical domain, AI finds widespread use in medical imaging research, encompassing improvements in image quality and the segmentation of tumor and healthy region boundaries to aid in surgical procedures [73,74,75]. Quantifying image information transforms disease diagnosis from qualitative to quantitative [76]. The use of AI to predict disease progression and treatment outcomes and even extend its analytical capabilities to wearable devices is an active area of research [77]. Furthermore, AI plays a pivotal role in designing and optimizing nanoparticles for drug delivery systems, enhancing drug targeting specificity [78]. In the realm of precision medicine, AI algorithms leverage patient-specific characteristics, lifestyle habits, and disease conditions to provide personalized treatment plans [79]. The functionality of AI extends to assisting in personalized healthcare, offering tailored solutions based on the algorithmic analysis of individual patient information.

## 4. CFD Models

Computational Fluid Dynamics (CFD) is widely employed for the description of fluid flow, multiphase flows, heat and mass transfer, and combustion phenomena. It finds application across various domains such as aerospace, automotive engineering, chemical engineering, energy systems, biomedicine, and more, encompassing a diverse range of problem types [80,81,82]. In recent years, CFD has also found utility in the medical field and SCF technology. Given that supercritical experiments involve high-temperature, high-pressure, closed-system processes akin to black-box operations, CFD enables the simulation of internal behaviors during experimental reactions on a computer, yielding visualized outcomes. Notably, it is used for simulating different operating conditions and nozzle geometries to explore the optimal mixing efficiency of organic solvents and SCCO_2_. The fundamental principles of CFD simulation are grounded in three major conservation equations: the continuity Equation (16), the momentum balance Equation (17), and the total energy balance Equation (18). The specific forms of these conservation equations may vary across different fluid systems, but a generalized representation of the conservation equations for CFD simulations in supercritical nanoparticle preparation typically appears as follows:

Continuity equation:(16)$$\frac{\partial \rho }{\partial t}+\nabla \cdot \left(\rho v\right)=0$$

Momentum balance equation:(17)$$\frac{\partial \rho v}{\partial t}+\nabla \cdot \left(\rho vv+\tau_{eff}\right)=\nabla p+\rho g$$

Total energy balance equation:(18)$$\frac{\partial \left(\rho E\right)}{\partial t}+\nabla \cdot \left(\rho vH+q_{eff}\right)=\nabla \left(\tau_{eff}\cdot v\right)$$
where ρ is the density of the fluid, *t* is time, *v* is velocity, $\tau_{eff}$ is the effective tensor, g is the gravity vector, *E* is the total energy, *H* is the total enthalpy, and $q_{eff}$ is an effective heat flux.

The implementation of a CFD model implies the selection of tools such as ANSYS Fluent (ANSYS, Inc., Canonsburg, PA, USA) [83], a software renowned for its expertise in fluid analysis, or the open-source CFD software package OpenFOAM (OpenCFD Ltd., Manchester, UK) [84]. As depicted in Figure 6, the fundamental workflow for simulating SCF technology using ANSYS Fluent is illustrated. Initially, it is essential to perform three-dimensional modeling of the experimental apparatus on the computer, followed by the generation of a grid on this three-dimensional model. This grid serves the purpose of facilitating computations. Subsequently, the solver is to be developed, encompassing tasks such as model selection, parameter configuration, establishment of boundary conditions, initialization, and other pertinent aspects. Ultimately, the calculation process is executed to obtain converged results.

For supercritical experiments, CFD enables the investigation of factors such as temperature, pressure, initial fluid velocity, and the impact of experimental apparatus, providing a visual representation of the diffusion pattern of drug solutions within SCF. Given that SCF technology primarily focuses on the efficiency of mixing organic solvents with SCCO_2_, it generally does not involve complex heat transfer or swirling phenomena. Therefore, opting for the k-ε turbulence model not only reduces computational demands but also ensures a certain level of accuracy. The selection of the k-ε turbulence model (19) is sufficient to accurately simulate the supercritical particle formation process.
$$\frac{\partial k}{\partial t}+\nabla \cdot [\rho vk-\left(\mu +\frac{\mu_{t}}{\sigma_{t}}\right)\nabla k]=P-\rho \varepsilon$$
(19)$$\frac{\partial \rho \varepsilon }{\partial t}+\nabla \cdot \left[\rho v\varepsilon -\left(\mu +\frac{\mu_{t}}{\sigma_{\varepsilon }}\right)\nabla \varepsilon \right]=C_{\varepsilon 1}\frac{\varepsilon }{k}P-C_{\varepsilon 2}\frac{{\rho \varepsilon }^{2}}{k}$$
where *k* is turbulent kinetic energy, $\mu_{t}$ is the turbulent viscosity, ε is the rate of turbulent kinetic energy dissipation, and $\sigma_{t}$,$\;\sigma_{\varepsilon }$, $C_{\varepsilon 1},$ and $C_{\varepsilon 2}$ are the k-ε model constants.

Moreover, supercritical experiments exhibit exceptionally rapid reaction times, enabling CFD to investigate changes in the motion state on minute time scales. This approach allows for exploring optimal experimental operating conditions and refining experimental equipment while conserving resources in terms of materials and time. In the case of supercritical particle formation via the SAS process, the conditions conducive to producing microparticles, submicron particles, and nanoparticles arise from the drug’s high solubility in organic solvents and its low solubility in SCCO_2_. This leads to the phenomenon of solute nucleation due to a rapid decrease in drug solubility upon mixing the organic solution with SCCO_2_, resulting in supersaturation. The good miscibility of organic solvents with SCCO_2_, facilitated by swift mixing, promotes rapid nucleation. Furthermore, the flow rates of the organic solution and SCCO_2_ entering the supercritical apparatus can influence the particle morphology [85]. CFD is adept at describing and simulating fluid mixing, and it can be leveraged to obtain solid–fluid surface tension, a challenging experimental measurement. By employing CFD to simulate different surface tension values and comparing the simulated particle size distribution with the experimentally obtained particle size distribution, surface tension can be estimated. High fluid diffusivity and low surface tension are critical factors for achieving a high nucleation rate [86]. Within the realm of CFD, the User-Defined Function (UDF) feature is available, allowing for the addition and modification of existing models using C or C++. In simulating SCF technology, UDFs can be employed to describe changes in density with respect to temperature and pressure, the mixing of organic solvents with SCCO_2_, and more complex experimental processes [87].

CFD can also simulate particle trajectories in supercritical particle formation experiments, employing specific models such as the Discrete Element Method (DEM) and multiphase flow models (Euler–Euler and Euler–Lagrange). DEM focuses on interactions between particles and is suitable for situations with higher particle concentrations [88]. Multiphase flow models, on the other hand, address interactions between fluids and particles and are well-suited for scenarios with lower particle concentrations [89]. Within multiphase flow models, the Population Balance Model (PBM) has been employed to simulate supercritical particle formation technology. As depicted in Figure 7, Cardoso et al. utilized a CFD single-way coupled particle population balance Equation (PBE, 23) to describe particle distribution during the SAS process for particle production. This approach revealed the relationship between initial flow rates, particle size distribution, and particle density. At the nozzle outlet, where the organic solvent and SCCO_2_ initially mix due to high flow velocities, particle density is high and particle size is small [90]. Given that in supercritical particle formation, drug particles are relatively scarce compared to SCCO_2_, the impact of particle motion on the SCF can be disregarded. SCF phase information, on the other hand, provides positional data for the particles. Therefore, using a CFD single-way coupled PBE offers the advantage of reducing computational demands while maintaining precision.
(20)$$\frac{\partial n\left(d_{p};t\right)}{\partial t}+\nabla \cdot \left[vn\left(d_{p};t\right)\right]=-\nabla_{d_{p}}\cdot \left[Gn\left(d_{p};t\right)\right]$$
where *n* is the density of the population, *d_p_* is the size of the particles, and *G* is the growth rate of the particles.

In addition to this, it is possible to simulate the impact of different reaction vessels on experimental mixing efficiency using computer modeling. This includes varying nozzle sizes, altering shapes, and changing inlet positions, among other factors. By comparing simulated parameters such as particle size, turbulence intensity, velocity magnitude, flow patterns, and volume fractions of different phases, one can demonstrate improved mixing efficiency and thus select suitable experimental apparatus [22,91,92]. As illustrated in Figure 8, CFD simulations of the SAS for curcumin particle preparation using T-shaped and cross-shaped nozzles reveal the distinct effects on the experiment. CFD results can display the magnitude of turbulence within the nozzles, with cross-shaped nozzles exhibiting higher turbulence intensity at the convergence point of the organic solvent and SCCO_2_, indicating greater mixing efficiency (Figure 8(ib)). Figure 8ii shows the SEM images of the *C. mangga* particles resulting from the cross and T nozzles. By comparing the SEM images, it is evident that the particles produced by the cross nozzle are smaller and more spherical than those produced by the T nozzle. This approach of using CFD modeling to simulate experiments for optimizing experimental equipment not only reduces the consumption of experimental materials but also saves costs and time in manufacturing experimental setups.

Currently, the application of Computational Fluid Dynamics (CFD) in the research on SCF preparation of nanoparticles is relatively limited. In recent years, it has been frequently employed in studies involving supercritical circulating fluidized beds. Prior to the industrial-scale application of new boiler equipment, using CFD to test its actual state can reduce financial and human resource consumption. By employing the Euler–Lagrange method, CFD assesses the heating area of multiphase flow and particle behavior to test whether combustion is thorough, thereby enhancing boiler combustion efficiency and reducing fuel consumption [93]. Utilizing a CFD-DEM model to simulate the cyclone separator of a supercritical water circulating fluidized bed reactor can enhance both mechanisms and optimize performance when separating incompletely burned coal powder [94]. In the industrial context, SCO_2_ exhibits excellent heat transfer properties, and CFD simulation of sCO_2_ heat transfer behavior assists in the production of industrial equipment [95,96]. For the supercritical drying process of gel particles, CFD models can track changes in velocity, pressure, temperature, composition, and other parameter fields of the studied system under arbitrary boundaries and initial conditions, optimizing equipment to reduce drying time [97]. In the medical field, CFD is commonly used in vascular imaging to aid in precise medical treatments [98]. Ferrofluids, which are colloidal solutions of metal nanoparticles, are employed in cancer treatment through magnetic hyperthermia. CFD can assist in real-time particle tracking for this application [99]. Additionally, CFD can be utilized to study drug release behavior and design drug delivery schemes [100].

## 5. Challenges and Opportunities

The preparation of nanoparticles using SCF technology presents numerous challenges [101]. Modeling plays a crucial role in facilitating the reduction in drug particle size in SCF technology. Massias et al. leveraged the Chrastil model and PR-EoS to predict the solubility of Nifedipine (NIF) within SCCO_2_ [102]. This predictive approach was complemented by the calculation of saturation, supersaturation, and driving forces, all aimed at optimizing the process of reducing NIF particle size through the Rapid Expansion of Supercritical Solutions (RESS) technique. At a constant temperature of 353.15K, the computed driving forces under pressures of 15, 24, and 30 MPa were 3010.9, 4443.9, and 5528.1 KJ mol^−1^, respectively. Correspondingly, experimental particle sizes yielded values of 38, 25, and 9 µm (Figure 9b). It was observed that higher pressures resulted in larger driving forces (Figure 9a), signifying a faster nucleation rate compared to crystal growth, and consequently, the generation of smaller particles. In contrast, the influence of temperature on both driving force and particle size was relatively modest when compared to pressure (Figure 9b). Additionally, temperature primarily impacted solubility, with the solubility decreasing as the temperature increased below the crossover pressure of 20.6 MPa at 15 MPa, leading to an increase in particle size. Conversely, at pressures exceeding the crossover point (24 MPa and 30 MPa), an increase in temperature led to higher solubility and smaller particle sizes.

In this study, we refer to models that address problems by constructing mathematical equations as mathematical models. These models encompass empirical, thermodynamic, and molecular dynamics models. Mathematical models are highly accurate when simplifying and idealizing problems, making them amenable to validation and replication. They are also the most versatile, capable of computing solubility, density, viscosity, thermal conductivity, diffusivity, and other properties of SCFs. However, their drawback lies in the complexity of the equations, which can render computations challenging, particularly for thermodynamic and molecular dynamics models. Moreover, mathematical models often rely on idealized assumptions, limiting their ability to capture intricate phenomena. For intricate systems, simplifying the problem may result in an inaccurate depiction of system behavior. In the case of empirical and semi-empirical models, direct fitting to experimental data is a black-box process, constraining the understanding of the model’s internals and rendering them unsuitable for predicting solubility in complex multicomponent systems. Furthermore, their applicability is limited, requiring parameter refitting for different systems or even varying conditions within the same system. Choosing a semi-empirical model poses a challenge: models with fewer parameters yield suboptimal fitting, while those with more parameters increase computational complexity. Adjusting parameters is an empirical process, introducing the risk of overfitting or underfitting. In the context of supercritical granulation experiments, the process involves multiple components. Mixing rules are a crucial component of the state equation, enabling it to describe complex multicomponent systems effectively [103]. The development of empirical and semi-empirical methods for calculating the solubility of inorganic compounds in subcritical and supercritical water lags behind thermodynamic models used for phase equilibria [104]. However, obtaining the parameters and initial conditions necessary for the state equation typically demands a considerable amount of experimental data, including thermodynamic parameters like activity coefficients and enthalpy, making calculations intricate. Without a comprehensive thermodynamic theoretical foundation, understanding and further expansion become challenging. For mathematical models, parameter calculation and adjustment constitute the primary sources of error. Balancing between computational complexity and practical application corresponds to finding an equilibrium between theoretical and empirical aspects, presenting a significant challenge.

The shortcomings associated with mathematical models are particularly evident when considering the advantages of AI models. AI methods require only inputs and outputs, delegating the intermediate processes to computer algorithms. AI models excel in handling nonlinear relationships, allowing them to learn complex dependencies between temperature, pressure, and solubility from extensive experimental data. Additionally, AI models possess strong feature extraction capabilities, automatically identifying latent features. Consequently, they not only predict solubility but may also reveal other critical factors influencing solubility. This flexibility in parameter exploration is a notable asset of AI. For instance, when investigating the impact of varying depressurization times on particle size in supercritical particle formation, traditional mathematical models often require assumptions about the continuous depressurization affecting drug solubility in SCCO_2_. In contrast, AI models can simply incorporate depressurization time as an additional feature for training. However, it is worth noting that AI models come with their own set of limitations. They rely heavily on substantial and high-quality experimental data, as the accuracy of prediction models is greatly dependent on both the quantity and quality of the data available. AI models typically require a substantial amount of annotated data for training. If the data quality is poor, biased, or insufficient, it may limit the model’s performance, and its generalization ability in new environments or domains may be compromised. In the context of supercritical nano-drug preparation, collecting experimental data is challenging, and measurements carry a certain degree of error. Additionally, the dataset needs to encompass a wide and sufficient range of data; otherwise, AI models may struggle to accurately generalize to unseen samples [105]. Due to the difficulty in collecting a large volume of such experimental data, the AI algorithms employed are often confined to conventional ML models, making the application of large-scale deep learning models impractical. The application of AI in chemical reactions faces similar challenges, primarily due to the involvement of multiple molecules and their diverse interactions in the reaction environment, coupled with the scarcity of samples [106]. This limitation hinders its widespread application, and establishing a corresponding database presents a significant challenge.

If parameters for SCCO_2_ and the mixed solution can be obtained through literature searches or software tools, CFD can directly simulate the fluid. Alternatively, simulation particle sizes can be brought closer to experimental sizes through parameter assumptions. In addition, the advantages of CFD include generating visual representations of experimental processes, reducing experimental costs and time, serving as a substitute for experimental research, and aiding in equipment optimization. However, CFD has its limitations, especially when dealing with complex flow phenomena that require substantial computational resources. This is particularly evident in cases such as supercritical experiments, which involve high-temperature and high-pressure processes akin to black-box operations, lacking experimental validation and relying on empirical judgment. CFD analysis is widely applied in mechanical engineering, but it encounters numerous limitations in medical applications. On the one hand, CFD models typically rely on certain assumptions. While some of these assumptions are helpful in understanding problems, they introduce deviations between simulations and real-world scenarios. CFD serves as a primary method for hemodynamic analysis, necessitating assumptions such as non-Newtonian fluid properties and rigid vessel walls. However, these assumptions significantly deviate from reality, considering the substantial variations in blood and vessel characteristics among individuals, imposing limitations on the applicability of CFD analysis [107]. In the context of turbulent fluid flow, the choice of turbulence models can impact simulation results. However, no single model is universally applicable, requiring empirical judgment or considerable time for comparison. Due to the complexity and continuity of fluid flow, influenced by the geometry of the computational domain and the mesh, CFD often demands substantial computational resources, restricting the scale and complexity of simulations. For the preparation of nanoparticles using SCF technology, simulating the experimental process involves solid, liquid, and SCF phases. Considering different phase velocities and interactions, the computational cost of conducting three-phase simulations using CFD tools may be high [108]. This is often simplified to solid-liquid two-phase simulation. Crystallization, as a pivotal process, contributes to particle size enlargement not only through nucleation and crystal growth but also through direct interactions between particles. While CFD-PBM can simulate the rate, temperature, and mixing behavior of nanoparticle synthesis under different conditions, it cannot accurately predict particle aggregation, as revealed in SEM images [109]. Therefore, CFD can only offer a partial explanation in this regard.

Mathematical models serve as the foundation for both AI models and CFD models. Assuming computational power is not a limiting factor, mathematical models can effectively address the same problems that AI and CFD models seek to solve. The core of AI models relies on algorithms based on mathematical methods, logical reasoning, and data processing. The core of CFD models also revolves around solving equations. Bagheri et al. employed both mathematical and CFD models to simulate pressure, temperature, and density distributions at different nozzle positions in the RESS process, with both models exhibiting consistent trends [110]. When conducting CFD simulations, material properties must be provided. Common materials can often be found in the software’s built-in database. However, SCCO_2_ is generated under high-temperature and high-pressure conditions, mirroring the conditions of experimental reactions. Measuring material properties during experimental reactions under these extreme conditions can be costly. While ANSYS Fluent includes the NIST real gas model, which can describe the properties of SCCO_2_, it cannot capture the properties of mixed solutions [111]. Therefore, mathematical models are essential for calculating the density, viscosity, thermal conductivity, and diffusivity of SCCO_2_ and mixed solutions during experimental reactions. For instance, the PR equation can be used to calculate density [112,113]. AI and CFD models are valuable and efficient tools for addressing complex mathematical models, serving as supplements to mathematical modeling. Pure mathematical modeling has rarely been applied directly to study process parameters in SCF technology, and the results obtained may lack intuitiveness [114]. CFD, on the other hand, is well-suited for studying fluid flow, heat transfer processes during supercritical processes, and, especially, predicting particle morphology in supercritical particle formation experiments [115].

It is essential to note that choosing the appropriate model type is crucial for specific problems and applications. Table 7 summarizes the conditions required and the objectives to be achieved when using these models for the preparation of nanoparticles using the SCF technology. In the context of this work, AI models and CFD models address different aspects. AI methods excel at predicting solubility, while CFD focuses on optimizing experimental equipment and processes. When abundant experimental data, such as solubility and particle size distribution, are available, employing AI models for predicting solubility and particle size is a precise and rapid approach. In cases with limited data, empirical models can be utilized, and for a more comprehensive understanding, complex mathematical models may need to be constructed. When the emphasis is on improving experimental equipment and seeking suitable experimental conditions and data availability is limited, using CFD models becomes a preferable choice. In the field of SCF technology, especially supercritical particle formation, there are reliable thermodynamic models like the PR model for describing the thermodynamic properties of SCF.

However, AI models and CFD models for supercritical particle formation remain relatively scarce. These two approaches are crucial tools for elucidating the collective effects of drugs, organic solvents, and SCF in the particle formation process. Combining various models is also a viable option, such as using data obtained from CFD simulations as input features for ML models to augment data and further enhance predictive capabilities. In the realm of indoor environmental design, Computational Fluid Dynamics (CFD) can be employed to simulate and predict air distribution and pollutant dispersion. Utilizing AI methods for prediction enhances efficiency [116]. Additionally, CFD methods alone cannot discern the interrelationships among parameters such as velocity, temperature, and turbulence. Applying AI models to CFD results allows for learning these associations and variations [117]. The primary advantage of CFD-AI lies in its capability to enhance the predictive capacity of parameters and improve processing speed. CFD generates a significant volume of discretized equations and computational data at each step. Coupling each step with AI algorithms and using AI to compute differential equations and perform parameter optimization proves effective in enhancing result accuracy. Employing AI algorithms facilitates the generation of more efficient and rational grids, significantly reducing computational time [118]. AI algorithms can optimize the representation of complex phenomena in CFD, such as turbulence, multiphase flow, and chemical reactions. Babanezhad et al. employed CFD to predict the flow of nanofluids in porous pipes and utilized AI algorithms to learn CFD results for velocity prediction [119]. Lira et al. used CFD to obtain data unobservable and unmeasurable in experiments. Leveraging AI models on the basis of 256 CFD simulation data significantly improved the predictive capability to R^2^ = 0.9997 [120]. The supercritical granulation experimental process involves multiphase flow, and relying solely on CFD to explain this process is limited. Using a CFD-DEM model to obtain particle positions as input features for AI models allows for the estimation of individual resistances for each particle in the system [121]. In the research on the supercritical preparation of nanoparticles, the utilization of CFD-AI models is becoming a prominent trend as a crucial tool for investigating interphase interactions and crystal growth behavior.

## 6. Conclusions

This review provides a comprehensive analysis of the models applied in the preparation of nanoparticles using SCF technology, including mathematical models, AI models, and CFD models, yielding important insights into this field. Our review findings demonstrate that these models can assist in predicting experimental results, optimizing experimental equipment, and reducing experimental costs. During the review process, the following conclusions were drawn: Firstly, we discovered that different models play a crucial role in explaining and predicting experimental results in the preparation of nanoparticles using SCF technology. Through a comprehensive analysis of multiple studies, we revealed the applicability, advantages, and limitations of different models. These findings are significant for advancing the theoretical understanding of supercritical nanoparticle preparation and promoting the development of nanoscale drug particles. However, it is important to acknowledge the limitations of this review, particularly the scarcity of research that utilizes experimental results to validate the superiority of models. Based on the findings of this review, it is recommended that future research further combine different models, such as AI intelligence predictions into CFD models to reduce computational costs while increasing the volume of data. In summary, this review offers an important comprehensive analysis and insights into the models applied in the preparation of nanoparticles using SCF technology. We hope that these findings provide valuable references for future research and inspire further exploration and expansion in this field by scholars.

## Data Availability

No new data were created or analyzed in this study. Data sharing is not applicable to this article.

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
