# Peer review of "Simulation and Optimization: A New Direction in Supercritical Technology Based Nanomedicine"

_bioengineering, 2023, doi:10.3390/bioengineering10121404_

Round 1
Reviewer 1 Report
Comments and Suggestions for Authors
General Comments
This paper by Huang Y et al. presents a review encompassing the application of mathematical models, artificial intelligence (AI) methodologies, and computational fluid dynamics (CFD) techniques within supercritical fluids technology used for nanomedicine preparation.
I have a few concerns mainly regarding the manuscript’s format.
Define all abbreviations at their first appearance and then use only the acronym throughout the text, tables, or figure caption.
After all equations describe the meaning of variables e.g., “where p is the pressure, v is the velocity”.
Provide only the surname of first authors when citing using et al. (e.g., Huang et al.).
Specific Comments
Page 2, line 73: define PVP.
Page 3: section 2. Mathematical model should be 2. Mathematical models.
Provide units of measure for y in Figures 1-4.
Page 7, line 263: there is a missing author “et al. employed 30 empirical…”.
Page 12, line 327: starting “the” should be “The”.
Figure 4: there is no a) and b).
Page 14, line 395: define NIST REFPROP.
Page 16, lines 467, 469: write R2 with 2 as superscript, as in Table 4.
Page 17, line 486: “presently” should be “Presently”.
Page 20: NU-SVR should be NU-SVM.
Table 5: try to improve the caption, including the units of measure (kg m-3 should use -3 as superscript); consider using scientific notation for solubility, etc.
Page 21, line 529: “allows for” should be better “implies”; provide references for ANSYS Fluent and OpenFOAM.
Author Response
General Comments
- Define all abbreviations at their first appearance and then use only the acronym throughout the text, tables, or figure caption.
Authors’ response: We thank the reviewer for helpful comments, and we have made more detailed proofreading and corrected the existing issues in revised manuscript
- After all equations describe the meaning of variables e.g., “where p is the pressure, v is the velocity”.
Authors’ response: Thank you for your valuable feedback. We have carefully addressed your suggestion by providing detailed explanations of variables in all equations. Each equation now includes a description of the meaning of variables, as exemplified by 'where p represents pressure, v represents velocity.' Additionally, all symbols and units are appropriately annotated in Appendix A. We believe these modifications enhance the clarity and comprehensibility of the manuscript.
- Provide only the surnameof first authors when citing using et al. (g., Huang et al.).
Authors’ response: We thank the reviewer for helpful comments, and we have made more detailed proofreading and corrected the existing issues in revised manuscript
Specific Comments
- Page 2, line 73: define PVP.
Authors’ response: We thank the reviewer for helpful comments, and we have made more detailed proofreading and corrected the existing issues in revised manuscript.
- Page 3: section 2. Mathematical model should be 2. Mathematical models.
Authors’ response: We thank the reviewer for helpful comments, and we have made more detailed proofreading and corrected the existing issues in revised manuscript.
- Provide units of measure for y in Figures 1-4.
Authors’ response: Thank you for your valuable feedback. We have rectified the oversight regarding the unit for the mole fraction solubility (y) in both the text and figures. The unit "mol·mol-1" has been consistently applied throughout the manuscript.For detailed information, please refer to the Appendix A at the end of the manuscript.
- Page 7, line 263: there is a missing author “et al. employed 30 empirical…”.
Authors’ response: We thank the reviewer for helpful comments, and we have made more detailed proofreading and corrected the existing issues in revised manuscript.
- Page 12, line 327: starting “the” should be “The”.
Authors’ response: We thank the reviewer for helpful comments, and we have made more detailed proofreading and corrected the existing issues in revised manuscript.
- Figure 4: there is no a) and b).
Authors’ response: We thank the reviewer for helpful comments, and we have made more detailed proofreading and corrected the existing issues in revised manuscript.
- Page 14, line 395: define NIST REFPROP.
Authors’ response: Thank you for your valuable feedback. We apologize for any confusion in our previous submission regarding NIST REFPROP. In response to your suggestion, we have enhanced the clarity of our description by explicitly defining NIST REFPROP.
Modified part in revised manuscript:
“... and mixed solutions [57, 58]. The NIST REFPROP software program stands as a powerful tool for calculating the thermophysical properties of industrially significant fluids. Within its capabilities, numerous reliable models are incorporated, enabling the direct retrieval of properties such as density and viscosity for both pure fluids and their mixtures [59]. ”
References:
[59] Huber, Marcia L. and Lemmon, Eric W. and Bell, Ian H. and McLinden, Mark O. The NIST REFPROP Database for Highly Accurate Properties of Industrially Important Fluids. Industrial \& Engineering Chemistry Research 2022, 61, 15449-15472. https://doi.org/10.1021/acs.iecr.2c01427
- Page 16, lines 467, 469: write R2 with 2 as superscript, as in Table 4.
Authors’ response: We thank the reviewer for helpful comments, and we have made more detailed proofreading and corrected the existing issues in revised manuscript.
- Page 17, line 486: “presently” should be “Presently”.
Authors’ response: We thank the reviewer for helpful comments, and we have made more detailed proofreading and corrected the existing issues in revised manuscript.
- Page 20: NU-SVR should be NU-SVM.
Authors’ response: Thank you for your valuable feedback. We have carefully considered your suggestion and made the necessary adjustments in response. Specifically, we have corrected the abbreviation of the model in the references to "Nu-SVR." The full term and abbreviation first appear on page 17, line 467, and we have ensured consistency in the model description in the second column of the Table 6.
- Table 5: try to improve the caption, including the units of measure (kg m-3 should use -3 as superscript); consider using scientific notation for solubility, etc.
Authors’ response: We thank the reviewer for helpful comments, and we have made more detailed proofreading and corrected the existing issues in revised manuscript.
- Page 21, line 529: “allows for” should be better “implies”; provide references for ANSYS Fluent and OpenFOAM.
Authors’ response: Thank you for your valuable feedback. We appreciate your suggestion to replace "allows for" with "implies" in the manuscript. We have made the necessary revisions accordingly to enhance the clarity and precision of the text.Additionally, as requested, we are providing the references for ANSYS Fluent and OpenFOAM in the appropriate format:
Modified part in revised manuscript:
The implementation of a CFD model implies the selection of tools such as ANSYS Fluent [82], a software renowned for its expertise in fluid analysis, or the open-source CFD software package OpenFOAM [83].
References:
[82] ANSYS. ANSYS Fluent [Computer software]. (2023). https://www.ansys.com/products/fluids/ansys-fluent
[83] OpenFOAM Foundation. OpenFOAM - The Open Source CFD Toolbox [Computer software]. (2023). https://www.openfoam.com/
Reviewer 2 Report
Comments and Suggestions for Authors
This article reports on simulation models for supercritical technology. It would be worthy of publication if the following points were corrected.
The text in figures 2, 4 and 7 is too small to read. Please improve.
Regarding Table 1, why is the number of parameters different depending on the target drug? Is it because the analysis models used are different?
I think it is important to develop a simulation method that can reproduce actual phenomena, but a table such as 'This method is good for this type of system' would be of interest to readers.
It is well written. It is worthy of acceptance if the sections I commented on to the author can be corrected.
Author Response
- The text in figures 2, 4 and 7 is too small to read. Please improve.
Authors’ response:Thank you for bringing the issue regarding the text size in Figures 2, 4, and 7 to our attention. We have revised and improved the text size in these figures to ensure better readability. The updated figures have been incorporated into the manuscript. We appreciate your careful review and valuable feedback.
- Regarding Table 1, why is the number of parameters different depending on the target drug? Is it because the analysis models used are different?
Authors’ response: Thank you for your thoughtful observation regarding Table 1. The variation in the number of parameters is not attributed to differences in drugs but rather arises from variations in the employed models. The primary purpose of Table 1 is to summarize the frequency of usage of commonly employed empirical models in recent years. Additionally, listing the drug names facilitates quick reference for individuals seeking to optimize experiments using these models. From the perspective of model development, empirical models are crafted based on an array of existing experimental data, observing trends in data variations to formulate mathematical representations. When considering the selection of models for predicting solubility, each empirical model is capable of predicting drug solubility. Models with fewer parameters entail simpler calculations, while those with more parameters exhibit superior fitting capabilities. In this study, the referenced articles explore different parameter models for fitting the solubility of the same drug. The aim is to aid in the selection of an appropriate model or to substantiate conclusions such as "models with more parameters yield higher accuracy." Therefore, there is no inherent connection between the choice of drug and the model; rather, it is a strategic decision based on the goals of the investigation.
- I think it is important to develop a simulation method that can reproduce actual phenomena, but a table such as 'This method is good for this type of system' would be of interest to readers.
Authors’ response: Thank you for your thoughtful comments and suggestions. We agree with your emphasis on the importance of developing a simulation method that can reproduce actual phenomena. In response to your suggestion, we have incorporated Table 7 into the manuscript. This table is designed to assist readers in selecting an appropriate simulation method by highlighting its suitability for different types of systems. Each entry in the table provides guidance on when the method is particularly effective, offering readers valuable insights into its applicability.We believe that the inclusion of Table 7 significantly enhances the manuscript, providing a concise and reader-friendly reference for choosing the most suitable simulation method. We hope this addresses your concern and adds value to the overall content.
Modified part in revised manuscript:
“...problems and applications. Table 7 summarizes the conditions required and the objectives to be achieved when using these models for the preparation of nanoparticles using the SCF technology. ”
Reviewer 3 Report
Comments and Suggestions for Authors
In this article, the authors have reviewed various methods for predicting the production of nanomedicines using supercritical fluid technology. The review extensively covers four distinct approaches, namely empirical models, models based on physical state modeling, artificial intelligence modeling, and numerical simulation methods. The tables within the paper offer a comprehensive summary of the development and interconnections among different articles, aiding the audience in identifying the model best suited for their applications.
However, there are areas where the manuscript could be enhanced:
- Introduction of Supercritical Fluid Technology: While the authors mention the application of nanoparticles synthesized through supercritical fluid technology as nanomedicine, the manuscript lacks a clear distinction between particles produced via this method and those from alternative synthesis approaches. It would be beneficial for the authors to delve into a discussion about the unique physical and chemical properties of materials generated through supercritical fluid technology and how these properties can be applied to address health-related issues.
- Characterizations of Nanoparticles: The absence of experimental results until Figure 8 may hinder the audience's understanding of the technology and the significance of simulations. To enhance comprehension, consider introducing experimental results earlier in the manuscript. Connecting these results with the simulation parameters discussed in various models, especially by indicating key parameters in schematics, would provide a clearer link between theory and practice for the readers. This approach can enhance the overall coherence of the paper and help the audience grasp the practical implications of the simulation models discussed.
English is fine.
Author Response
- Introduction of Supercritical Fluid Technology: While the authors mention the application of nanoparticles synthesized through supercritical fluid technology as nanomedicine, the manuscript lacks a clear distinction between particles produced via this method and those from alternative synthesis approaches. It would be beneficial for the authors to delve into a discussion about the unique physical and chemical properties of materials generated through supercritical fluid technology and how these properties can be applied to address health-related issues.
Authors’ response: Thank you for your insightful feedback. In the introduction section, we discussed the benefits of supercritical fluid technology in the preparation of nanomedicine. Compared to conventional techniques like precipitation, emulsion, and spray drying, they have limitations such as the inclusion of organic solvents and the propensity to render thermosensitive medications inactive. The solvent SCCO2 is non-toxic and easily removed, which lowers the risk of hazardous organic solvent residues and safeguards the bioactivity and structural integrity of some biomacromolecules that are sensitive to heat. In addition, it offers accurate control over the drug's particle size, morphology, and crystal structure, which facilitates the simple manufacturing of pure pharmaceuticals in batches. We have also presented the work done by our team to improve the imaging performance of fluorescence navigation and the effectiveness of interventional embolization treatment for liver cancer, which is based on supercritical fluid technology. In conclusion, the SCF approach offers a solid platform for the development of drug delivery systems and pharmaceutical particles with potential for clinical application.
- Characterizations of Nanoparticles: The absence of experimental results until Figure 8 may hinder the audience's understanding of the technology and the significance of simulations. To enhance comprehension, consider introducing experimental results earlier in the manuscript. Connecting these results with the simulation parameters discussed in various models, especially by indicating key parameters in schematics, would provide a clearer link between theory and practice for the readers. This approach can enhance the overall coherence of the paper and help the audience grasp the practical implications of the simulation models discussed.
Authors’ response: Thank you for your insightful feedback. We appreciate your attention to the experimental validation aspect of our manuscript. Throughout the main text, specific examples of empirical models, equation of state (EOS) models, and artificial intelligence (AI) models have been provided, illustrating the synergy of experimental data and model representation. The first five figures, which correspond to the sections on mathematical models and AI models, depict the specific models that were developed by fitting parameters to experimental solubility data. These models were then compared to the data for validation, and the figures present the shared information between the experiments and the models. Figure 1 depicts the experimental solubility data obtained at different pressures and temperatures. In Figures 2, 3, and 4, the scattered data points represent the experimentally measured solubility, while the curves represent the models fitted to the data. Placing them together in the same graph allows for a visual comparison. Similarly, in Figure 5, a set of scattered data points represents the difference between the experimental solubility and the measured solubility. The excellence of the models in capturing experimental trends is evident through the numerical values of indicators such as R2 and AARD%. In the revised version, we have enhanced the discussion by including specific numerical comparisons, further strengthening the argument.However, concerning Computational Fluid Dynamics (CFD) models, we have primarily presented examples of simulation results. This choice is attributed to the fact that CFD is a tool commonly employed for engineering problems, allowing the depiction of parameters that are challenging to obtain through experiments. We acknowledge and highlight in the manuscript that one limitation of CFD is the lack of experimental validation, emphasizing the need to rely on theoretical knowledge and experience to enhance the accuracy of CFD models during construction.We trust that these modifications address your concerns, and we remain open to further suggestions.
Modified part in revised manuscript:
“...(AARD%) was employed. AARD% is a standard for assessing the disparity between experimental and calculated values. A lower AARD% indicates a smaller deviation between the model and experimental data. Among the six models considered, the Keshmiri model (vi) exhibits the lowest AARD% = 10.630, signifying the highest fitting accuracy [24]. ”
“...various EoS models. The correlation indicators R2 for both models exceed 0.9, indicating that these models can explain over 90% of the variability. Hence, the dissolution of chloroquine in SCCO2 can be predicted using these models. ”
“...the actual values. Based on the correlation coefficient R2 between the models and experimental data, the R2 values for the DT model, ADA-DT model, and Nu-SVR model are 0.836, 0.921, and 0.813, respectively. Consequently, the ADA-DT model, with an R2 exceeding 92.1%, demonstrates the highest explanatory power and is deemed the most representative of the experimental data, while the Nu-SVR model exhibits the lowest precision. ”
Reviewer 4 Report
Comments and Suggestions for Authors
The manuscript delves into the application of mathematical models, artificial intelligence (AI) methodologies, and computational fluid dynamics (CFD) techniques in the realm of supercritical technology for nanomedicine preparation. It addresses the challenges arising from the unique conditions of supercriticality, such as the difficulty in directly measuring parameters like drug solubility, temperature, pressure, and mixing efficiency. The authors extensively review methodologies for calculating drug solubility in supercritical fluids (SCFs) and assess the impact of operational conditions and experimental apparatus on nanomedicine outcomes. While showcasing the merits and demerits of commonly used models, the manuscript advocates for the reliability of employing models to compute drug solubility and simulate experimental processes. It highlights the practical implications of these models in aiding and optimizing experiments, as well as guiding the selection of operational conditions, fostering innovative avenues for the development of supercritical pharmaceuticals. However, potential weaknesses include a limited discussion on the models' limitations, the use of complex technical language that might hinder accessibility for a broader audience, and the manuscript's potential lack of the latest developments in the rapidly evolving field. The manuscript can be recommended for publication subject to the following revisions.
1. Table 5 can be rewritten using 10^? To avoid the long number of decimal points in outputs.
2. The manuscript briefly mentions limitations of mathematical models, AI, and CFD. A more thorough discussion on the limitations, challenges, and potential sources of errors associated with each model would strengthen the manuscript.
3. The manuscript lacks explicit references to recent advancements or applications of AI and CFD in nanomedicine or supercritical technology. Including recent examples or studies would enhance the relevance and up-to-dateness of the review.
4. While the manuscript discusses AI and CFD models independently, exploring potential synergies or integrations between these approaches could provide valuable insights into their combined use for enhanced predictive capabilities.
5. The manuscript briefly touches on experimental validation but lacks in-depth discussion or examples. Providing specific cases or studies where models were validated against experimental data would strengthen the manuscript's credibility.
6. The introduction can be improved and and field of nanomedicine need to be better explained. Some recommendation include:
Nanomaterials Theory and Applications. Encyclopedia of Smart Materials, Volume 3, 2022, Pages 302-314.
Recent advances in Cu(II)/Cu(I)-MOFs based nano-platforms for developing new nano-medicines. Journal of Inorganic Biochemistry, Volume 225, December 2021, 111599.
Author Response
- Table 5 can be rewritten using 10^? To avoid the long number of decimal points in outputs.
Authors’ response: We thank the reviewer for helpful comments, and we have made more detailed proofreading and corrected the existing issues in revised manuscript.
- The manuscript briefly mentions limitations of mathematical models, AI, and CFD. A more thorough discussion on the limitations, challenges, and potential sources of errors associated with each model would strengthen the manuscript.
Authors’ response: We thank the reviewer for the valuable suggestion and we have added relevant content.
Modified part in revised manuscript:
“...capture intricate phenomena. For intricate systems, simplifying the problem may result in an inaccurate depiction of system behavior. In the case of empirical and semi-empirical models, direct fitting to experimental data is a black-box process, constraining the understanding of the model's internals and rendering them unsuitable for predicting solubility in complex multicomponent systems. Furthermore, their applicability is limited, requiring parameter refitting for different systems or even varying conditions within the same system. Choosing a semi-empirical model poses a challenge: models with fewer parameters yield suboptimal fitting, while those with more parameters increase computational complexity. Adjusting parameters is an empirical process, introducing the risk of overfitting or underfitting. In the context of supercritical granulation experiments, the process involves multiple components. Mixing rules are a crucial component of the state equation, enabling it to describe complex multicomponent systems effectively [103]. The development of empirical and semi-empirical methods for calculating the solubility of inorganic compounds in subcritical and supercritical water lags behind thermodynamic models used for phase equilibria [104]. However, obtaining the parameters and initial conditions necessary for the state equation typically demands a considerable amount of experimental data, including thermodynamic parameters like activity coefficients and enthalpy, making calculations intricate. Without a comprehensive thermodynamic theoretical foundation, understanding and further expansion become challenging. For mathematical models, parameter calculation and adjustment constitute the primary sources of error. Balancing between computational complexity and practical application corresponds to finding equilibrium be-tween theoretical and empirical aspects, presenting a significant challenge.”
“...the data available. AI models typically require a substantial amount of annotated data for training. If the data quality is poor, biased, or insufficient, it may limit the model's performance, and its generalization ability in new environments or domains may be compromised. In the context of supercritical nanodrug preparation, collecting experimental data is challenging, and measurements carry a certain degree of error. Additionally, the dataset needs to encompass a wide and sufficient range of data; otherwise, AI models may struggle to accurately generalize to unseen samples [105]. Due to the difficulty in collecting a large volume of such experimental data, the AI algorithms employed are often confined to conventional machine learning models, making the application of large-scale deep learning models impractical. The application of AI in chemical reactions faces similar challenges, primarily due to the involvement of multiple molecules and their diverse interactions in the reaction environment, coupled with the scarcity of samples [106]. This limitation hinders its widespread application, and establishing a corresponding database presents a significant challenge.”
“...on empirical judgment. CFD analysis is widely applied in mechanical engineering, but it encounters numerous limitations in medical applications. On the one hand, CFD models typically rely on certain assumptions. While some of these assumptions are helpful in understanding problems, they introduce deviations between simulations and real world scenarios. CFD serves as a primary method for hemodynamic analysis, necessitating assumptions such as non-Newtonian fluid properties and rigid vessel walls. However, these assumptions significantly deviate from reality, considering the considerable variations in blood and vessel characteristics among individuals, imposing limitations on the applicability of CFD analysis [107]. In the context of turbulent fluid flow, the choice of turbulence models can impact simulation results. However, no single model is universally applicable, requiring empirical judgment or considerable time for comparison. Due to the complexity and continuity of fluid flow, influenced by the geometry of the computational domain and the mesh, CFD often demands substantial computational resources, restricting the scale and complexity of simulations. For the preparation of nanoparticles using supercritical technology, simulating the experimental process involves solid, liquid, and supercritical phases. Considering different phase velocities and interactions, the computational cost of conducting three-phase simulations using CFD tools may be high [108]. This is often simplified to solid-liquid two-phase simulation. Crystallization, as a pivotal process, contributes to particle size enlargement not only through nucleation and crystal growth but also through direct interactions between particles. While CFD-PBM can simulate the rate, temperature, and mixing behavior of nanoparticle synthesis under different conditions, it cannot accurately predict particle aggregation, as revealed in SEM images [109]. Therefore, CFD can only offer a partial explanation in this regard.”
References:
- Kenneth S. Pitzer. Thermodynamics of electrolytes. I. Theoretical basis and general equations. The Journal of Physical Chemistry1973 77 (2), 268-277.
https://pubs.acs.org/doi/10.1021/j100621a026
- Qinli Liu, Xin Ding, Bowen Du & Tao Fang. Multi-Phase Equilibrium and Solubilities of Aromatic Compounds and Inorganic Compounds in Sub- and Supercritical Water: A Review, Critical Reviews in Analytical Chemistry2017,47:6, 513-523.
https://doi.org/10.1080/10408347.2017.1342528
[105] Nateghi H, Sodeifian G, Razmimanesh F, Mohebbi Najm Abad J. A machine learning approach for thermodynamic modeling of the statically measured solubility of nilotinib hydrochloride monohydrate (anti-cancer drug) in supercritical CO2. Sci Rep 2023,13(1):12906. https://doi.org/10.1038/s41598-023-40231-4.
[106] Shilpa S, Kashyap G, Sunoj R B. Recent Applications of Machine Learning in Molecular Property and Chemical Reaction Outcome Predictions. The Journal of Physical Chemistry A 2023, 127(40): 8253-8271. https://doi.org/10.1021/acs.jpca.3c04779
[107] Cho K C. The Current Limitations and Advanced Analysis of Hemodynamic Study of Cerebral Aneurysms. Neurointervention 2023, 18(2): 107-113.
https://doi.org/10.5469/neuroint.2023.00164.
[108] D'Bastiani C, Kennedy D, Reynolds A. CFD Simulation of Anaerobic Granular Sludge Reactors: A Review. Water Research 2023: 120220.https://doi.org/10.1016/j.watres.2023.120220.
[109] Li, Q.; Wang, Z.; Wang, X. CFD–PBM Simulation for Continuous Hydrothermal Flow Synthesis of Zirconia Nanoparticles in a Confined Impinging Jet Reactor. Materials 2023, 16, 3421. https://doi.org/10.3390/ma16093421.
- The manuscript lacks explicit references to recent advancements or applications ofAI and CFD in nanomedicine or supercritical technology. Including recent examples or studies would enhance the relevance and up-to-dateness of the review.
Authors’ response: We thank the reviewer for the valuable suggestion and we have added relevant content.
Modified part in revised manuscript:
“...in this context [68-70].
In recent years, significant progress has been made in the fields of AI-SCF technology and nanomedicine, owing to the substantial advancements in computer hardware that have greatly enhanced computational capabilities. In the realm of super-critical technology, AI models are not only applied to predict drug solubility but also assist in optimizing experimental design and parameters [71]. In the medical domain, AI finds widespread use in medical imaging research, encompassing improvements in image quality, segmentation of tumor and healthy region boundaries to aid in surgical procedures [72-74]. Quantifying image information transforms disease diagnosis from qualitative to quantitative [75]. The use of AI to predict disease progression, treatment outcomes, and even extending its analytical capabilities to wearable devices is an active area of re-search [76]. Furthermore, AI plays a pivotal role in designing and optimizing nanoparticles for drug delivery systems, enhancing drug targeting specificity [77]. In the realm of precision medicine, AI algorithms leverage patient specific characteristics, lifestyle habits, and disease conditions to provide personalized treatment plans [78]. The functionality of AI extends to assisting in personalized healthcare, offering tailored solutions based on the algorithmic analysis of individual patient information.”
“...manufacturing experimental setups.
Currently, the application of CFD in the research on SCF)preparation of nanoparticles is relatively limited. In recent years, it has been frequently employed in studies involving supercritical circulating fluidized beds. Prior to the industrial-scale application of new boiler equipment, using CFD to test its actual state can reduce financial and human resource consumption. By employing the Euler-Lagrange method, CFD assesses the heating area of multiphase flow and particle behavior to test whether combustion is thorough, thereby enhancing boiler combustion efficiency and reducing fuel consumption [93]. Utilizing a CFD-DEM model to simulate the cyclone separator of a supercritical water circulating fluidized bed reactor can enhance both mechanisms and optimize performance when separating incompletely burned coal powder [94]. In the industrial context, SCCO2 exhibits excellent heat transfer properties, and CFD simulation of SCCO2 heat transfer behavior assists in the production of industrial equipment [95,96]. For the supercritical drying process of gel particles, CFD models can track changes in velocity, pressure, temperature, composition, and other parameter fields of the studied system under arbitrary boundaries and initial conditions, optimizing equipment to reduce drying time [97]. In the medical field, CFD is commonly used in vascular imaging to aid in precise medical treatments [98]. Ferrofluids, which are colloidal solutions of metal nanoparticles, are employed in cancer treatment through magnetic hyperthermia. CFD can assist in realtime particle tracking for this application [99]. Additionally, CFD can be utilized to study drug release behavior and design drug delivery schemes [100].”
References:
- Ahmad J. Obaidullah. Advanced AI modeling and optimization for determination of pharmaceutical solubility in supercritical processing for production of nanosized drug particles, Case Studies in Thermal Engineering,2023,49,103199.
https://doi.org/10.1016/j.csite.2023.103199.
- Nayak, A., Baidya Kayal, E., Arya, M. et al. Computer-aided diagnosis of cirrhosis and hepatocellular carcinoma using multi-phase abdomen CT. Int J CARS2019,14, 1341–1352.
https://doi.org/10.1007/s11548-019-01991-5.
[73] Haiyan Zheng, Yufei Chen, Xiaodong Yue, Chao Ma, Xianhui Liu, Panpan Yang, Jianping Lu. Deep pancreas segmentation with uncertain regions of shadowed sets,Magnetic Resonance Imaging 2020,68, 45-52.https://doi.org/10.1016/j.mri.2020.01.008.
[74] Hectors, S.J., Kennedy, P., Huang, KH. et al. Fully automated prediction of liver fibrosis using deep learning analysis of gadoxetic acid–enhanced MRI. Eur Radiol 2021,31, 3805–3814.
https://doi.org/10.1007/s00330-020-07475-4.
[75] Bi, W.L.; Hosny, A.; Schabath, M.B.; Giger, M.L.; Birkbak, N.J.; Mehrtash, A.; Allison, T.; Arnaout, O.; Abbosh, C.; Dunn, I.F. Artificial intelligence in cancer imaging: Clinical challenges and applications. CA A Cancer J. Clin. 2019, 69, 127-157.https://doi.org/10.3322/caac.21552.
[76] Kadirvelu B, Gavriel C, Nageshwaran S, Chan JPK, Nethisinghe S, Athanasopoulos S, Ricotti V, Voit T, Giunti P, Festenstein R, Faisal AA. A wearable motion capture suit and machine learning predict disease progression in Friedreich's ataxia. Nat Med 2023,29, 86-94.
https://doi.org/10.1038/s41591-022-02159-6.
[77] Govindan, B.; Sabri, M.A.; Hai, A.; Banat, F.; Haija, M.A. A Review of Advanced Multifunctional Magnetic Nanostructures for Cancer Diagnosis and Therapy Integrated into an Artificial Intelligence Approach. Pharmaceutics 2023, 15, 868.
https://doi.org/10.3390/pharmaceutics15030868.
[78] Mukhopadhyay, A.; Sumner, J.; Ling, L.H.; Quek, R.H.C.; Tan, A.T.H.; Teng, G.G.; Seetharaman, S.K.; Gollamudi, S.P.K.; Ho, D.; Motani, M. Personalised Dosing Using the CURATE.AI Algorithm: Protocol for a Feasibility Study in Patients with Hypertension and Type II Diabetes Mellitus. Int. J. Environ. Res. Public Health 2022, 19, 8979. https://doi.org/10.3390/ijerph19158979.
[93] Ying Cui, Wenqi Zhong, Xuejiao Liu, Jun Xiang. Study on scale-up characteristics in supercritical CO2 circulating fluidized bed boiler by 3D CFD simulation, Powder Technology 2021,394. https://doi.org/10.1016/j.powtec.2021.08.028.
[94] Zeyu Li, Zhenbo Tong, Hao Zhang, Kaiwei Chu, Renjie Li, Hao Miao, Jiansong Zhao, Aibing Yu, CFD-DEM simulation of the supercritical water-solid flow in cyclone,Powder Technology, 2023,418:118261.https://doi.org/10.1016/j.powtec.2023.118261.
[95] Anjun Li, Fernando Hernández Jiménez, Eduardo Cano Pleite, Zhenbo Wang, Liyun Zhu. Numerical comparison of thermal energy performance between spouted, fluidized and fixed beds using supercritical CO2 as fluidizing agent, Case Studies in Thermal Engineering 2022,39:102469, https://doi.org/10.1016/j.csite.2022.102469.
[96] A.E. Lebedev, D.D. Lovskaya, N.V. Menshutina, Modeling and scale-up of supercritical fluid processes. Part II: Supercritical drying of gel particles, The Journal of Supercritical Fluids 2021, 174:105238,https://doi.org/10.1016/j.supflu.2021.105238.
[97] Jianyong Wang, Jishuang Gong, Xin Kang, Chunrong Zhao, Kamel Hooman,Assessment of RANS turbulence models on predicting supercritical heat transfer in highly buoyant horizontal flows,Case Studies in Thermal Engineering 2022,34:102057.
https://doi.org/10.1016/j.csite.2022.102057.
- Ethan Winkler, David Wu, Eugene Gil, David McCoy, Kazim Narsinh, Zhengda Sun, Kerstin Mueller, Jayden Ross, Helen Kim, Shantel Weinsheimer, Mitchel Berger, Tomasz Nowakowski, Daniel Lim, Adib Abla, Daniel Cooke, Endoluminal Biopsy for Molecular Profiling of Human Brain Vascular Malformations. Neurology 2022, 98 (16) e1637-e1647;
https://doi.org/10.1212/WNL.0000000000200109.
[99] Farooq, U., Hassan, A., Fatima, N. et al. A computational fluid dynamics analysis on Fe3O4–H2O based nanofluid axisymmetric flow over a rotating disk with heat transfer enhancement. Sci Rep 2023,13, 4679. https://doi.org/10.1038/s41598-023-31734-1.
[100] Henriquez, F.; Celentano, D.; Vega, M.; Pincheira, G.; Morales-Ferreiro, J.O. Modeling of Microneedle Arrays in Transdermal Drug Delivery Applications. Pharmaceutics 2023, 15, 358. https://doi.org/10.3390/pharmaceutics15020358.
- While the manuscript discusses AI and CFD models independently, exploring potential synergies or integrations between these approaches could provide valuable insights into their combined use for enhanced predictive capabilities.
Authors’ response: We thank the reviewer for the valuable suggestion and we have added relevant content.
Modified part in revised manuscript:
“...enhance predictive capabilities. In the realm of indoor environmental design, Computational Fluid Dynamics (CFD) can be employed to simulate and predict air distribution and pollutant dispersion. Utilizing AI methods for prediction enhances efficiency [116]. Additionally, CFD methods alone cannot discern the interrelationships among parameters such as velocity, temperature, and turbulence. Applying AI models to CFD results allows for learning these associations and variations [117]. The primary advantage of CFD-AI lies in its capability to enhance the predictive capacity of parameters and improve processing speed. CFD generates a significant volume of discretized equations and computational data at each step. Coupling each step with AI algorithms, using AI to compute differential equations and perform parameter optimization, proves effective in enhancing result ac-curacy. Employing AI algorithms facilitates the generation of more efficient and rational grids, significantly reducing computational time [118]. AI algorithms can optimize the representation of complex phenomena in CFD, such as turbulence, multiphase flow, and chemical reactions. Babanezhad et al. employed CFD to predict the flow of nanofluids in porous pipes and utilized artificial intelligence algorithms to learn CFD results for velocity prediction [119]. Lira et al. used CFD to obtain data unobservable and unmeasurable in experiments. Leveraging AI models on the basis of 256 CFD simulation data significantly improved predictive capability to R²=0.9997 [120]. The super-critical granulation experimental process involves multiphase flow, and relying solely on CFD to explain this process is limited. Using a CFD-DEM model to obtain particle positions as input features for AI models allows for the estimation of individual resistances for each particle in the system [121]. In the research on the supercritical preparation of nanoparticles, the utilization of CFD-AI models is becoming a prominent trend as a crucial tool for investigating interphase interactions and crystal growth behavior.”
References:
[116] Development of self-adaptive low-dimension ventilation models using OpenFOAM: Towards the application of AI based on CFD data. Building and Environment 2020 171,106671. https://doi.org/10.1016/j.buildenv.2020.106671
[117] Babanezhad, M., Behroyan, I., Taghvaie Nakhjiri, A. et al. Prediction of turbulence eddy dissipation of water flow in a heated metal foam tube. Sci Rep 2020,10, 19280.
https://doi.org/10.1038/s41598-020-76260-6.
[118] Bo Wang and Jingtao Wang. Application of Artificial Intelligence in Computational Fluid Dynamics,Industrial & Engineering Chemistry Research 2021 60 (7), 2772-2790. https://doi.org/10.1021/acs.iecr.0c05045.
[119] Babanezhad, M., Behroyan, I., Marjani, A. et al. Velocity prediction of nanofluid in a heated porous pipe: DEFIS learning of CFD results. Sci Rep 2021,11, 1209.
https://doi.org/10.1038/s41598-020-79913-8
[120] Lira J O B, Riella H G, Padoin N, et al. Computational fluid dynamics (CFD), artificial neural network (ANN) and genetic algorithm (GA) as a hybrid method for the analysis and optimization of micro-photocatalytic reactors: NOx abatement as a case study. Chemical Engineering Journal 2022, 431: 133771. https://doi.org/10.1016/j.cej.2021.133771.
[121] Long He, Danesh K. Tafti, A supervised machine learning approach for predicting variable drag forces on spherical particles in suspension, Powder Technology 2019, 345:379-389. https://doi.org/10.1016/j.powtec.2019.01.013.
- The manuscript briefly touches on experimental validation but lacks in-depth discussion or examples. Providing specific cases or studies where models were validated against experimental data would strengthen the manuscript's credibility.
Authors’ response:Thank you for your insightful feedback. We appreciate your attention to the experimental validation aspect of our manuscript.Throughout the main text, specific examples of empirical models, equation of state (EOS) models, and artificial intelligence (AI) models have been provided, illustrating the synergy of experimental data and model representation. The excellence of the models in capturing experimental trends is evident through the numerical values of indicators such as R2 and AARD%. In the revised version, we have enhanced the discussion by including specific numerical comparisons, further strengthening the argument.However, concerning Computational Fluid Dynamics (CFD) models, we have primarily presented examples of simulation results. This choice is attributed to the fact that CFD is a tool commonly employed for engineering problems, allowing the depiction of parameters that are challenging to obtain through experiments. We acknowledge and highlight in the manuscript that one limitation of CFD is the lack of experimental validation, emphasizing the need to rely on theoretical knowledge and experience to enhance the accuracy of CFD models during construction.We trust that these modifications address your concerns, and we remain open to further suggestions.
Modified part in revised manuscript:
“...(AARD%) was employed. AARD% is a standard for assessing the disparity between experimental and calculated values. A lower AARD% indicates a smaller deviation between the model and experimental data. Among the six models considered, the Keshmiri model (vi) exhibits the lowest AARD% = 10.630, signifying the highest fitting accuracy [24]. ”
“...various EoS models. The correlation indicators R2 for both models exceed 0.9, indicating that these models can explain over 90% of the variability. Hence, the dissolution of chloroquine in SCCO2 can be predicted using these models. ”
“...the actual values. Based on the correlation coefficient R2 between the models and experimental data, the R2 values for the DT model, ADA-DT model, and Nu-SVR model are 0.836, 0.921, and 0.813, respectively. Consequently, the ADA-DT model, with an R2 exceeding 92.1%, demonstrates the highest explanatory power and is deemed the most representative of the experimental data, while the Nu-SVR model exhibits the lowest precision. ”
- The introduction can be improved and and field of nanomedicine need to be better explained. Some recommendation include:
Nanomaterials Theory and Applications. Encyclopedia of Smart Materials, Volume 3, 2022, Pages 302-314.
Recent advances in Cu(II)/Cu(I)-MOFs based nano-platforms for developing new nano-medicines. Journal of Inorganic Biochemistry, Volume 225, December 2021, 111599.
Authors’ response: We appreciate this important comment. In the revised manuscript, we have included the definition, characteristics, and advantages of nanomedicine at the beginning of the introduction.
Modified part in revised manuscript:
“ According to current definitions, nanomedicines are nanoscale tools utilized for dis-ease diagnosis, prevention, and treatment [1-2]. They have the capacity to facilitate early disease detection and prevention, direct a bioactive molecule to the intended site of action, and regulate the release of the molecule to guarantee an ideal concentration at the target of therapy for the intended duration [3-4].
Numerous nanomedicines, including liposomes, metal-organic frameworks (MOFs), polymeric nanosystems, magnetic nanoparticles, offer advantages over traditional thera-pies that generally alter the pharmacokinetics and pharmacodynamics (pK/pD) of the ac-tive ingredients [5-6]. However, there are still a number of issues with traditional na-nosizing techniques including precipitation, emulsion, and spray-drying that need to be resolved [7].”
References:
- Fan D, Cao Y, Cao M, Wang Y, Cao Y, Gong T. Nanomedicine in cancer therapy. Signal Transduct Target Ther. 2023,8(1):293. https://doi.org/10.1038/s41392-023-01536-y.
- Cheng J, Huang H, Chen Y, Wu R. Nanomedicine for Diagnosis and Treatment of Atherosclerosis. Adv Sci (Weinh). 2023, 2304294. https://doi.org/10.1002/advs.202304294.
- Ortiz-Pérez A, Zhang M, Fitzpatrick LW, Izquierdo-Lozano C, Albertazzi L. Advanced optical imaging for the rational design of nanomedicines. Adv Drug Deliv Rev. 2023, 115138. https://doi.org/10.1016/j.addr.2023.115138.
- Mengmeng Qin and Heming Xia and Wenhao Xu and Binlong Chen and Yiguang Wang. The spatiotemporal journey of nanomedicines in solid tumors on their therapeutic efficacy. Advanced drug delivery reviews2023: 115137. https://api.semanticscholar.org/CorpusID:265120338.
- Puri S, Mazza M, Roy G, England RM, Zhou L, Nourian S, Anand Subramony J. Evolution of nanomedicine formulations for targeted delivery and controlled release. Adv Drug Deliv Rev. 2023. 200:114962. https://doi.org/10.1016/j.addr.2023.114962.
- Fornaguera, C.; García-Celma, M.J. Personalized Nanomedicine: A Revolution at the Nanoscale. Pers. Med.2017, 7, 12. https://doi.org/10.3390/jpm7040012.
- Zhang M, Dai Z, Theivendran S, et al. Nanotechnology enabled reactive species regulation in biosystems for boosting cancer immunotherapy. Nano Today, 2021, 36: 101035.
https://doi.org/10.1016/j.nantod.2020.101035.
Round 2
Reviewer 1 Report
Comments and Suggestions for Authors
The authors responded satisfactorily to my suggestions, I have no further comments.
Author Response
Thanks for your kind help.